# High-Resolution Assessment of Air Quality in Urban Areas—A Business Model Perspective

**Klaus Schäfer** [1,*], **Kristian Lande** [2], **Hans Grimm** [1], **Guido Jenniskens** [3], **Roel Gijsbers** [3], **Volker Ziegler** [4], **Marcus Hank** [4] **and Matthias Budde** [5]

1    Aerosol Akademie e.V., 83404 Ainring, Germany; vorstand@aerosol-akademie.de
2    AirVeraCity, EPFL, 1015 Lausanne, Switzerland; kristian.lande@airveracity.com
3    ENS Clean Air, 5431 DH Cuijk, The Netherlands; g.jenniskens@ens-cleanair.com (G.J.);
     r.gijsbers@ens-cleanair.com (R.G.)
4    GRIMM Aerosol Technik Ainring GmbH & Co. KG, 83404 Ainring, Germany;
     Volker.Ziegler@grimm.durag.com (V.Z.); mhank@web.de (M.H.)
5    Chair for Pervasive Computing Systems/TECO, Karlsruhe Institute of Technology,
     76131 Karlsruhe, Germany; matthias.budde@disy.net
*    Correspondence: schaefer@atmosphericphysics.de; Tel.: +49-175-629-6313

**Abstract:** The increasing availability of low-cost air quality sensors has led to novel sensing approaches. Distributed networks of low-cost sensors, together with data fusion and analytics, have enabled unprecedented, spatiotemporal resolution when observing the urban atmosphere. Several projects have demonstrated the potential of different approaches for high-resolution measurement networks ranging from static, low-cost sensor networks over vehicular and airborne sensing to crowdsourced measurements as well as ranging from a research-based operation to citizen science. Yet, sustaining the operation of such low-cost air quality sensor networks remains challenging because of the lack of regulatory support and the lack of an organizational framework linking these measurements to the official air quality network. This paper discusses the logical inclusion of lower-cost air quality sensors into the existing air quality network via a dynamic field calibration process, the resulting sustainable business models, and how this expansion can be self-funded.

**Keywords:** low-cost sensor; middle-cost sensor; air pollutant; health protection; citizen science; sustainability

## 1. Introduction

If the first modern, disruptive, technological revolution was the Internet, in which people were connected in unprecedented ways, the next revolution, in which billions of devices are connected, has the potential to be much larger and holds the possibility of fundamentally changing how we interact with our environment. It is estimated that there could be up to 42 billion connected Internet of Things (IoT) devices by 2025 [1]. This IoT revolution—digitizing the physical world—has received enormous attention and, in combination with the increasing availability of low-cost air quality sensors (LCS), is leading to novel air quality sensing approaches [2].

These new approaches offer the greatest potential benefit in urban areas where often invisible, hyperlocal air pollution can vary by more than eight times within 200 m [3]. This type of personal, high-resolution air quality information is not available via the existing, official air monitoring networks whose purpose is primarily to measure regulatory pollutant limit violations. It was this collection of facts that led the former Mayor for New York, Michael Bloomberg, to declare that "collecting data from all sources of pollution, and making it publicly available, is one of the least expensive and most effective ways to clean the air" [4]. Furthermore, the EU Joint Research Centre, which is the research organization that provides independent scientific advice to support EU policy, said that LCS have the

potential to be a "game changer" in terms of understanding the environment provided that the quality of their measurements is improved [5].

The challenge for LCS is that they are trying to mimic highly accurate, sophisticated, and expensive reference-level air quality monitors. These existing reference-level monitors are used by the air quality authorities to fulfil regulatory and compliance requirements that are highly accurate (>90%), expensive (~$180 k per station), resemble laboratory equipment, and require specialist technical expertise to operate and maintain. LCS, on the other hand, tend to resemble small electromechanical devices, are relatively inexpensive (<$8 k per replicated station), easy to use, and are targeted to fulfil the role of indicative-level monitors.

In air quality assessment terms, an indicative measure is part of a commonly recognized and legislatively defined set of accuracy thresholds set for each pollutant. The legislatively defined threshold accuracies range from very stringent for regulatory purposes (>90%) to less demanding "indicative measurements" (>~70% depending on pollutant and assessment need) [6,7], which are achievable with lower cost sensors [8].

Accuracy is the primary obstacle for air quality authorities to use LCS at greater levels. All air quality monitoring equipment is significantly influenced by weather and drift. The weather is site-specific and the result of cross-sensitivities from interfering compounds on measurement performance [9]. However, drift, which is the loss of accuracy over time, varies uniquely across each device. To address these issues, well-known methods for quality assurance and control are applied to the laboratory equipment, such as the implementation of calibration materials and gases, by highly trained technicians. These techniques are not practical, and, in many cases, not possible with LCS, so they are dependent upon the existing air quality monitoring equipment to mimic.

This key problem of deployed device accuracy is recognized by many technical bodies such as the one in Europe, which is currently drafting a certification for LCS [10] and strongly recommends that sensor measurements are periodically compared side-by-side with the reference stations near the deployed location. This well-established field calibration process involves various statistical techniques and is critical for these LCS devices to achieve acceptable levels of accuracy or, according to the Council of Gas Detection and Environmental Monitoring, "actual measurements taken are only as good as the calibration of the instrument used to do the measuring" [11]. This means that any future certification will have limited practice relative to the algorithms used and frequency of the instruments' field calibration.

A key to the future business prospects of using LCS in air quality assessments depend on the existing air quality authorities' active support and involvement in their use. In a joint statement of 38 influential organizations from 14 different countries, a lack of cooperation between the official air quality monitoring networks (AQM), operators of the reference stations, and lower-cost sensor operators was identified as the main obstacle to their greater use for officially conducted air quality assessments [12].

This paper provides an overview, including some examples, of the existing and future business opportunities using LCS to conduct a high-resolution assessment of air quality in urban areas. It reviews the business cases for this under three scenarios: (1) the current situation in which LCS are operated independently of the AQM, (2) the current situation where greater air quality assessment needs are met through an integration of LCS into the official air quality network, and (3) a proposed way forward to systematically network LCS to the AQM using a dynamic field calibration process as well as the resulting benefits and business opportunities.

## 2. Background of Existing Business Models for High-Resolution Assessment of Air Quality

Air quality assessments have a wide range of accuracy target thresholds. These move beyond the legislative definition of providing indicative or regulatory compliance measures, as seen below [7]:

- Regulatory compliance (>90%)

- Spatial gradient studies (>75%)
- Intervention studies/indicative measures (>70%)
- Hot spot determination (>50%)
- Citizen science projects (>50%).

The status quo, in most areas of the world, is little or no cooperation between the official air monitoring networks and the lower-cost sensor operators. This situation relegates the use of LCS largely to citizen science projects and/or hobbyists. It is commonly assumed that the accuracy of these devices is not much better than 50%.

The most successful business model for urban air quality monitoring is the one-time sales of hardware and software. Today, there are hundreds of LCS systems commercially available on the market with costs ranging from several hundred to several thousand euros [9]. The total size of the global air quality monitoring market, which includes both reference level monitors and LCS hardware and services, is expected to reach $6.4 billion in the next 5 years [13]. The segment of this market experiencing the greatest growth are LCS hardware and services, which is growing at 14.3% [14] and the municipal IoT enabled LCS hardware to expand at an astounding 25.4% [12] per annum.

The reason that this market has been largely limited to hobbyists is because the current business model offers no guarantee of deployed sensor accuracy or precision. Nevertheless, the demand for LCS continues to grow, which is reflected in both the sales data and the changing public perception of their environment. A recent public opinion survey on air pollution in Europe [15] found that the majority of Europeans (54%) did not feel well-informed about air quality and, despite significant improvements [16], believed that air quality was getting worse.

LCS offer a feasible and affordable means to quantify the effect of air pollution mitigation interventions. High-resolution Smart Air Quality Networks (SAQN, see: Section 4) can be installed in the urban environment, parallel to policy-based or technology-based mitigation strategies, which may be integrated in the existing urban infrastructure ('Lungs of the City'-concept [17]). Encouraging the active participation of local hobbyists, residents' organizations, and educational institutions may co-create a 'living lab,' resulting in increased public awareness and sense of involvement. Availability of real-time air quality information among peers will increase acceptance of such data. Moreover, visualization of actual air quality improvements upon mitigation will positively affect public opinion, as well as the sentiment that changes can be taken to improve local air quality.

*2.1. Standalone, Independently-Operated LCS Air Quality Assessment Networks*

The demand for greater air quality information and a call for governments to provide this has led to numerous government-funded citizen science projects. These projects are undertaken by members of the general public, often in collaboration with or under the direction of professional scientists and scientific institutions [18].

The demand for this information has been so great that some citizen science projects have continued to grow even after the official end of the project. The most notable has been "Luftdaten", which started as a nationally funded Open Knowledge Lab project in Germany with the aim of making particulate matter (PM) visible in places where it was not officially measured [19]. Beginning in 2015 with 300 sensors in Stuttgart, Germany [20], it has now grown to a global network of over 12,168 privately owned LCS reporting over 7 billion data points and operating in 75 different countries. This post-project growth occurred without government funding and without an advertising budget. Luftdaten, now known as a sensor community (https://sensor.community/en/ (accessed on 2 May 2021), has plans for further expansion and wants to use its grassroots organization to pressure local governments to take further action to address air pollution [21].

*2.2. Semi-Integrated LCS-Official Air Quality Assessment Networks*

The general technological trajectory for LCS is clearly one of ever improving capability since newer sensors tend to outperform older versions [22]. This continually improving

performance, increasing consumer demand, and air quality authority's obligation to inform the public has naturally led these devices to become increasingly integrated into the existing AQM. This integration together with data fusion and analytics has allowed these devices to provide legislatively recognized indicative levels of accuracy, which has opened up many new applications, uses, and business opportunities.

The following real-world examples demonstrate different types of integration and the resulting benefits of an expanded, indicative air quality monitoring network.

### 2.2.1. AirNow: Fire and Smoke Map

In the United States, the Environmental Protection Agency (EPA) is currently advancing lower cost and portable air measurement technology to enhance monitoring capabilities that comply with the National Ambient Air Quality Standards (https://www.epa.gov/naaqs (accessed on 2 May 2021)) [23]. They have also taken the additional step of integrating an existing private LCS network (>10 k LCSs) operated by PurpleAir (https://www2.purpleair.com/ (accessed on 2 May 2021)) into a government-sponsored platform known as AirNow. The AirNow Fire and Smoke map is a sensor data pilot project designed to provide the public with additional information on particle pollution levels ($PM_{2.5}$) in the air, particularly during wildfires. This is provided via a web-based real-time map (https://fire.airnow.gov/ (accessed on 2 May 2021)) that fuses particle pollution measures from the official EPA air quality monitors, corrected PurpleAir LCS measurements, and fire data from the U.S. Forest Service.

In order to ensure the integrity of the LCS readings, the EPA applies a correction factor based on 50 collocated PurpleAir sensors situated at 39 sites spanning 16 different states (https://www.epa.gov/air-sensor-toolbox/technical-approaches-sensor-data-airnow-fire-and-smoke-map (accessed on 2 May 2021)) [24]. These correction factors help to account for systematic biases in the raw data measurements reported from each LCS, including biases caused by different atmospheric conditions. The release of this integrated product corresponded to one of the worst fire seasons in the Western US. Previously, AirNow received tens of thousands of page views during the entirety of a single smoke event. With the new integrated map, they reached a peak of nearly 400,000 page views on a single day, achieving over 7.4 million views between 14 August 2020 and 30 November 2020.

### 2.2.2. Cangzhou, China

In Cangzhou, China, a pilot project sponsored by the Environmental Defense Fund (EDF), Beijing Huanding Environmental Big Data Institute, and the municipal government was launched, showcasing the ability of mobile LCS technology to reduce air pollution by targeting enforcement actions in hot spot areas [25]. This pilot consisted of 50 taxis mounted with LCS measuring particulate matter every 3 s. Both mobile and stationary LCS measures were used to fill the gaps left by the official AQM network while relying on that network to ensure accuracy.

Within 3 months after starting this pilot, air pollution enforcement effectiveness increased by a factor of 10. This pilot was the result of a wider "grid management" project initiated in 2017 by the Ministry of Ecology and Environment, China, which led to a drop of PM2.5 levels by 38.3% in the piloted area versus a 20% reduction experienced across the rest of Cangzhou [26].

### 2.2.3. Rijnmond: We-Nose Network

As the level of integration and analysis with the official air quality stations increases, so does the usefulness of the data. Electronic noses (E-Noses) are a technology that have been around for more than 25 years and attempt to mimic mammalian smell. Historically, they had been deployed for quality control of food and other products as well as odour exposure in the surroundings of sources, such as from agriculture. However, in the last 10 years, they have increasingly been deployed as alert systems to detect ambient odour concentrations within ports, specifically for the illegal release of Volatile Organic Compounds (VOC).

These releases often occur during the degasification of liquid-carrying vessels but can also result from unintended leaks [27]. These E-Noses are generally deployed as a collection of non-specific, low-cost, gas-sensor arrays that, when properly trained and distributed, can identify the presence and source of the emissions.

In order to achieve this goal in an outdoor setting, rigorous field calibrations need to be completed as does extensive training of the data [28,29]. This process entails a mathematical pattern recognition of smells (both odour and odourless) as each type of substance has a unique chemical signature [30]. In practice, this means a very close cooperation with the local air quality authority figure who operates the nearby AQM stations where some of the E-Noses are located. When an event is detected and investigated, it often entails a comparison of the E-Nose measures and samples taken from traditional approaches conducted by the local air quality authority.

The Port of Rotterdam, situated in a densely populated region of the Netherlands, is the largest refining and chemical cluster in Europe. In 2013, as a result of numerous odour complaints (5–6 k per year), 77 E-Nose sensors were installed by a company to act as an early alert system to identify the location, concentration, and composition of these emissions. This network, known as We-Nose (https://www.dcmr.nl/projecten/e-nose-programma-rijnmond.html (accessed on 2 May 2021)), has now expanded to more than 250 E-Noses covering the Rijnmond area. The success of this integrated network has led to similar efforts in other ports, e.g., Amsterdam, Antwerp, and Tallinn. This type of application for ambient air quality is additive and not a replacement of the existing efforts. The triumph of this approach would not be possible without the combined collaborative efforts of the port, national environment agency, and private industry.

## 3. Potential Benefits of High-Resolution Assessments of Air Quality

Air pollution is the fifth leading risk factor for mortality worldwide and, each year, more people die from it than road traffic accidents or malaria [31]. In Europe, air pollution is the number one cause of premature deaths from environmental factors and this effect is particularly pronounced in cities where most people live [32]. In economic terms, the resulting loss due to urban air pollution is currently estimated at 3.9% of all income earned in cities and this cost "will probably increase as additional associations between pollution and disease are identified" [33].

### 3.1. Cost Savings Potential of Improved Public Health

The rapidly growing healthcare expenditures now exceed 10% of global GDP [34]. One clear payment strategy is to finance the expansion of the measurement networks with the money that is saved through reduced healthcare cost (Figure 1). The rationale for such a self-funding mechanism is as follows: Initially, policy-based or technology-based intervention and mitigation strategies (as designed and implemented by (local) governments and funded from public sources (€1)), result in improvements of AQ. As a consequence of appropriate governance-based actions (Figure 1, grey-shaded box), a positive effect on public health is anticipated, which, in turn, results in reductions of healthcare costs. Resources that are saved due to lower healthcare spending can be reassigned (€2) to intensify air pollution mitigation efforts and to expand monitoring networks. In turn, this results in additional AQ improvement, leading to further exposure reduction, and, consequently, additional health gain. Thus, an adequate governance framework may establish proper conditions to generate a self-sustainable 'positive feedback loop.'

In this scheme, AQM serves multiple purposes. Improved public awareness may prompt individuals to use actual measurement data in making informed decisions regarding personal exposure-evasion strategies. Additionally, the availability of current and past AQ information will greatly assist in determining causality, drawing up legislation, and implementing enforcement.

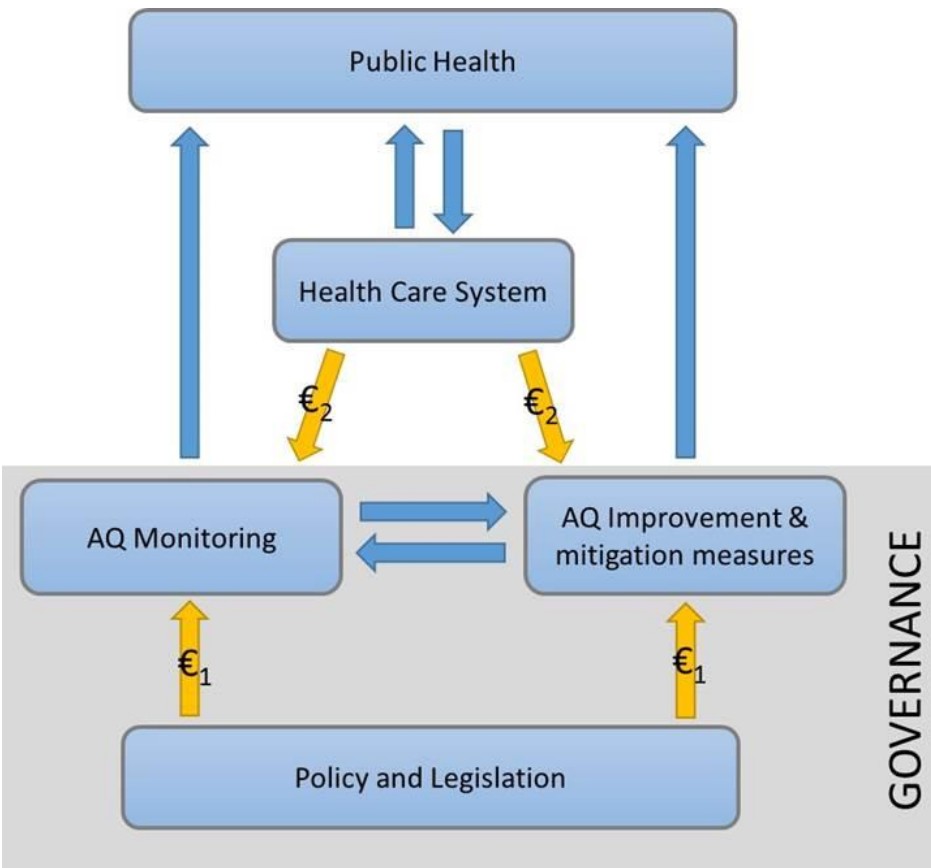

**Figure 1.** The interplay of air quality (AQ) monitoring and AQ improvement and mitigation measures, paid out of common assets (€1), adds to improved public health. The necessary measurement networks and intervention strategies could, therefore, be funded from the savings that were generated in the healthcare system (€2), gradually phasing out the necessity of a contribution from common assets.

Another topic for financing the costs of clean air is the willingness of people to pay for air quality data.

The advent of affordable, high-resolution air quality measurement networks allows for the reliable assessment of personal exposure potential and related health risk. Based on this type of information, individuals can make informed decisions regarding personal evasion strategies [35]. The ability to avoid urban air pollution concentrations using real-time measurements is not currently possible with the existing official network. This network's reference stations are used primarily for compliance purposes and typically go through a quality control process prior to public reporting. This reporting delay is typically between 30–120 min. Moreover, data are representative for the conditions at that specific station location only.

In Europe, a recent study focusing on urban air pollution concentrations concluded that the societal costs (see Figure 1: Public Health) for the 432 cities studied, totalled EUR 166 billion in 2018 [36]. These encompass the overwhelming majority of the European GDP [37]. To measure this exposure, the 891 urban reference stations in Europe had an operational cost of only EUR 21.3 million [16,37]. Unfortunately, the academic literature is very limited on the ability of people to avoid localized air pollution concentrations because these real-time measurements are not currently available. However, assuming that they were avoiding localized air pollution and this were to lead to only a 3% reduction in exposure in Europe, this would translate into a savings of approximately EUR 5 billion per year.

The highest estimates for the social cost of air pollution in Europe are in London. In 2018, the loss in welfare for its 8.8 million inhabitants totalled EUR 11.38 billion [36].

A similar 3% reduction in air pollution exposure would potentially translate into an economic benefit of EUR 341.4 million per year. For this reason, the City of London embarked on a multi-year consortium project focused on high-resolution air quality assessments. The first phase of this project involved 250 schoolchildren carrying LCS devices to measure their personal exposure to air pollution on their route to school. This phase also involved one-off mobile mapping of the city using two normally powered Google Street View cars (https://insights.sustainability.google/labs/airquality (accessed on 2 May 2021)). These were specifically fitted with reference grade air quality monitoring equipment and were driven on weekdays over the course of a year, using a previously developed process [38]. Upon completion of this mapping, 100 LCS were installed at air pollution hotspots (https://www.london.gov.uk/what-we-do/environment/pollution-and-air-quality/london-air-quality-map (accessed on 2 May 2021)). In order to verify the accuracy of those devices, they were periodically manually calibrated with reference level equipment.

The success of this project has led to its further extension until 2024. The extension involves fixed deployments by a company of 100 LCS at hospitals, schools, and other sensitive locations (https://www.breathelondon.org/ (accessed on 2 May 2021)) at a cost of roughly EUR 214,000 per year [39]. This figure is aligned with a general guide for the total costs of an LCS network in which the analysis expenditure is five times the hardware costs [40]. In the case of London, the benefit is very clear. If the deployment of LCS leads to a reduction of exposure by a mere 1%, simply through avoidance, the societal benefit could total EUR 113.8 million at a cost of only EUR 214,000 (a 531.7× return).

### 3.2. Polluter-Pays-Principle

The polluter-pays-principle is the basics of all environmental protection measures. However, the practical realisation is limited in the case of road, ship, or air traffic. High-resolution air pollution networks of stationary and mobile sensors can support the detection of unknown emission hot spots. This is an additional driving factor when developing such networks by agencies, which are responsible for compliance with the polluter-pays-principle. Following this, the knowledge of personal air pollution exposure and, thus, health risks is not only a personal interest but a societal interest.

### 4. Future: The Smart Air Quality Network

This paper outlines a standardized approach for the expansion of the existing air quality monitoring (AQM) network to include IoT-linked LCS devices, which we refer to as the Smart Air Quality Network (SAQN). This expanded indicative monitoring network is made possible through telecommunications, GPS positioning, and a mobile network of air pollution sensors mounted to normally operated vehicles. The critical link in this network occurs when two air pollution monitoring devices sample the same air. In an outdoor setting, this means that they pass within close proximity of each other. When this occurs, an automated comparison analysis of these air quality measurements is made, either within the LCS device or in the cloud, and calibration instructions are relayed back to the device in need of adjustment. The end result of this fully connected network will be the accuracy of every connected device in real-time. These measures can then be used for specific air quality assessments and to reduce the public's exposure to urban air pollution concentrations. Further adaptation of additional digitized information will be critical for some assessment needs and will improve the accuracy and reliability of the overall network.

The market for LCS sales will not be the focus of this paper since those sales are expected to continue double-digit growth into the foreseeable future, which is independent of establishing an organizing framework. However, given growing popularity, it should be noted that customers for LCS have shifted significantly in recent years from expert users to consumers. A lack of public understanding regarding the complexity of urban air pollution and trending negative public opinions about the trustworthiness of governments are growing risks that can ultimately undermine existing efforts if nothing is done. An

overview of low-cost sensors for particulate matter and gases in different monitors is listed together with references, tests conducted, the standard method used, the comparison period, and the outcome in Reference [41]. An overview about data communication, storage, cloud services, processing, and dissemination is given in Reference [41] as these areas are not discussed here.

### 4.1. The Smart Air Quality Network Design

The ability to link devices via sampling the same air in an outdoor setting is a common practice for all LCS devices and is referred to as a field calibration. The ability to do this on a mobile basis was successfully demonstrated in the Opensense project (http://opensense.epfl.ch/wiki/index.php/OpenSense_2.html (accessed on 2 May 2021)), which involved LCS mounted to 10 normally operated municipal buses. These buses were run continuously over the course of 3 years (600,000 driven km, 800 m data points). The result was these findings solved for the mobility distortion effects on slow reacting chemical sensors and they developed an automated, dynamic calibration process [42], which improved sensor accuracy and solved for drift. As in other studies, the vehicle emissions themselves did not significantly affect the LCS readings [43].

The Opensense project demonstrated that a network of mobile sensors can be consistently and reliably maintained remotely with very high levels of availability (>99%). While there is very little information available on performance comparisons using non-stationary sensor networks, this project demonstrated accuracy levels consistently above the existing indicative thresholds. In a comprehensive review of LCS performance based on stationary deployments conducted in 2019, sensors built using the same design and calibration techniques developed by Opensense were found to be the most accurate in the market at that time [9].

The future Smart Air Quality Network should be based on a similar dynamic calibration technique. A dynamic recalibration process is one that can take place systematically in real-time between any two devices. The instructions for the adjustment of a device that has been deemed to need adjustment are then sent back to that device.

The backbone of this chain of linked devices is the official air quality reference station, which must be in close proximity to at least one of the passing vehicles with mounted LCS devices. A hierarchy of accuracy and reliability is then established for the entire chain of linked devices, which is based on sampling frequency and number of devices, or nodes, between the device and the reference station, e.g., a single LCS that passes often within close proximity to the reference station would have a higher rank than a device that is several steps removed from passing a reference station and is now in a different pollution environment. For cities that have multiple reference stations where a single LCS device passes, there could be multiple sets of calibrations depending on a combination of location, weather variables, and pollution levels. As in any classic network, the information of this network improves with a greater number of nodes on the network, i.e., it follows Metcalf's Law, which states that the value of the network is proportional to the square of the number of users (see Figure 2).

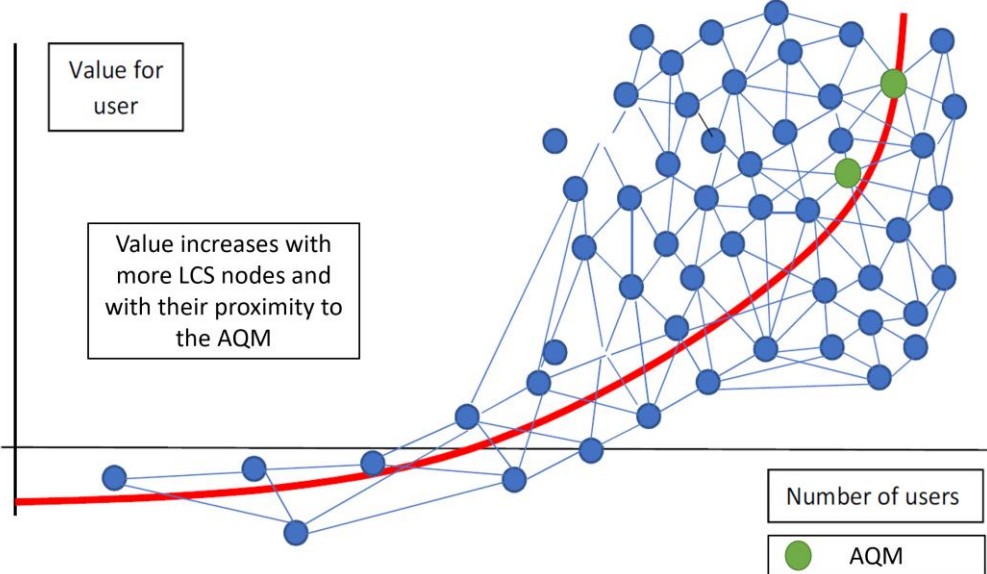

**Figure 2.** The value of the Smart Air Quality Network is proportional to both the square of the number of connected LCS devices and their spatial and temporal closeness to the existing official air quality monitoring network (AQM). These AQM stations are shown in green in the chart above.

For stationary devices that do not fit within this synchronized measurement chain, i.e., one that never passes another device, a different technique has been developed to calibrate devices remotely based on nearby measurements [2]. This was demonstrated in Beijing using 1000 LCS (including seven that were collocated with reference stations) [44]. These calibration instructions can then be transferred across a larger scale gestalt network of similar devices using the 5G network's advances in data processing and 3D positioning [2].

For areas not measured within a city, spatial pollution distribution maps, incorporating urban topography, have been developed using these high-frequency, spatially distributed measurements. One notable example is the method developed [45] as a result of a mobile sensing project in Antwerp (https://www.imeccityofthings.be/en/projects/dencity-more-sensors-in-the-city (accessed on 2 May 2021)) in which 20 postal vehicles were equipped with LCS and combined with the measurements from 15 fixed LCS locations.

The end results are real-time air pollution measurements available in 25-m increments across roughly 70% of the city. These are achieved with only a small number of mobile sensors (https://business.edf.org/insights/future-fleets/ (accessed on 2 May 2021)) coupled to the existing official network. These are then supplemented with additional measurements taken by interconnected personal LCS devices (both fixed and mobile) and newly developed dispersion modelling applying all these data points.

This approach has three additional significant benefits:

- It is technology independent as it can continue to grow and improve with newer sensing technologies,
- It is hyper scalable. The more nodes it has, the better it functions, as most of the work is done via algorithms and telecommunication technology,
- It allows the air quality authority to freely use sensors of a known accuracy that are owned and maintained by others without sacrificing the quality of its existing measurements.

### 4.2. Structuring and Funding the Smart Air Quality Network

The following section outlines how the expansion of the network can be self-funding and operated in a way that is additive to the air quality authority's assessment needs.

### 4.2.1. Mobile Network Operator

A business opportunity exists for an independent company to maintain the mobile network on behalf of the air quality authority. In the public's mind, this delineation more clearly separates the regulatory responsibilities of the air quality authority from those involving indicative measures. This delineation also makes functional sense as LCS are focused on information technology rather than environmental science.

### 4.2.2. LCS Franchise Fees

Each independent LCS owner would pay a fee to receive a certification from the environmental protection agency, i.e., their device meets the indicative measures quality threshold. The fee generation could be used to offset the network's operating cost. This franchise fee system could be further segmented by enforcement rights or a visual display of the air quality at that location.

### 4.2.3. LCS Franchise Fees: Green Halo

Many people want to be seen as "green." The subset of air pollution sensor owners known as the "Green Halo" pay an annual franchise fee to be visible on the network and seen on the official air quality maps. In addition, their sensor can be equipped with visual cues such as the LED lights on their vehicle that represents the current air quality level at that location. The concept of the "Green Halo" is originally a label given to the premium that people were willing to pay for a hybrid vehicle (~6%) and is based on the economic idea of conspicuous conservation [46].

### *4.3. Business Models*

The following section illustrates some examples of businesses that could be developed based on the existence of a newly expanded network.

### 4.3.1. Visibility: Hyper-Local Air Pollution Maps/Visual Cues

Visibility leads to a greater simplicity and understanding. This digital visualization of the real world can take the form of visual signals of localized air pollution, such as a colour-coded air quality index represented by an LED light. These lights could be placed at a street level to indicate the air quality at that location and encourage certain forms of transport.

Mobile phone applications based on higher levels of spatial and temporal data already exist, e.g., "Green Pathways" [47] and allow users to minimize their exposure, e.g., during their daily commute. These apps draw the community closer together since they have a similar appeal to other social media applications of this type, which fulfil people's need for purpose. The primary revenue drivers of this business model are advertising revenues.

### 4.3.2. Traffic Management: Flexible LEZs

There are around 250 Low Emission Zones (LEZ) [48] in Europe and it is unclear what their effectiveness is in reducing air pollution. As reported by the EU Court of Auditors, "in the rare instances when the effectiveness of these zones is assessed, it is often years after implementation" [32]. In addition, the fees collected very rarely cover the direct costs of administration. According to experts, low emissions zones do not make money using traditional enforcement mechanisms: "The more you enforce, the more expensive it gets" [49].

The integration of the SAQN data to the existing traffic control system allows these zones to be flexibly managed, which would increase their political acceptability and reduce the enforcement cost. The primary revenue drivers of this business model are derived from the government's transport budget.

### 4.3.3. City and Transport Planning

A key aspect of all planning processes is an assessment of the costs and benefits of the policy options being considered with a considerable focus on the 'cost benefit ratio' for the building or transport project [50]. Detailed guidelines on how government economists should quantify these effects is often very limited for air pollution. An SAQN would simplify and standardize this aspect of the process and allow verification of specific design effects on the environment, which is unprecedented. The primary revenue driver of this business model is local government funding.

### 4.3.4. Air Quality Plans

The previously mentioned auditor's report found that: "Existing air quality plans often do not assess the cost effectiveness of the measures taken, are insufficient, poorly targeted, and the enforcement process is slow" [32]. It maintained that citizens should "play a key role in monitoring the Member States' implementation of the Ambient Air Quality Directive, in particular, when results imply difficult political choices. Local action is important, but requires public awareness: only if citizens are well informed can they be involved in the policy and take action, where appropriate, including changing their own behaviour." The SAQN serves as a systematic, real-time check on the effectiveness of mitigation measures taken. The primary revenue driver of this business model is government funding.

### 4.3.5. City-Level Emissions Inventories

An emissions inventory is a database that lists, by source, the amount of air pollutants discharged into the atmosphere. Governments use emission inventories to help determine significant sources of air pollutants and to target regulatory actions [51].

The demand for city-level emission inventories is growing, with 9500 cities signed onto the Paris Agreement. Numerous cities have developed climate action or remediation plans, but only a small percentage have the necessary inventories to adequately track their progress [52,53].

Calculating an inventory for cities is a complex, expensive undertaking that requires access to accurate, citywide data, which is often not readily available or standardized. Essentially, these inventories are normally only done at a national level and are based on national statistics, e.g., fuel consumption. In an attempt to remedy this problem, standardized approaches have been developed [54] for cities and research has been done to further disaggregate this data spatially and temporally [55]. While these advances are necessary for complex aspects such as wood burning heat [56] or road salting [57], they inevitably rely on emission factors, which are assumed to represent long-term averages. The difficulty with this top-down approach has been most dramatically illustrated with the 'Dieselgate' scandal in which $CO_2$ emissions have been shown to be one-third higher [58] and $NO_X$ emissions up to 40 times higher [59] than official estimates. Even prior to this discovery, it was widely accepted that the uncertainty around emission factors for transport were regularly around 50% and rose to as high as 120% [60]. This uncertainty around city-level emissions using a top-down approach will only get worse with the electrification of the transport sector, which, in Europe, has witnessed a 212% increase since last year with the share of electric vehicles now reaching 10% of all new vehicle sales [61].

As previous research suggests, the approach using additional information to spatially and temporally disaggregate emissions data does yield big improvements and has been tried by companies like Google (https://insights.sustainability.google/ (accessed on 2 May 2021)). However, ultimately, the problem is that a top-down approach does not work well no matter how well the information is disaggregated because the correct and real emission indices must be known. What is required are a greater number of ambient air measurements for source apportionment and inverse dispersion modelling methods, i.e., a bottom-up approach [62]. This can be seen in the most recent academic review of 48 US cities in which it was found that they were, on average, under reporting their emissions by

18.3%, with a range of −145.5% to +63.5%. As a result, the authors of the report suggest a more systematic, bottom-up approach with spatial granularity down to the street level [63].

4.3.6. European Emissions Trading Scheme

One promising approach would be to incorporate the building and transport sectors into the existing European Emissions Trading Scheme (ETS). Since 2005, the ETS has included the power generation and industrial sectors, which account for 46% of GHG emissions and have been reduced by 25% since ETS inception. In contrast, the building and transport sectors, which account for 38% of GHG emissions, have only achieved marginal reductions in the same time period.

This works simply by combining the SAQN data with additional information, such as vehicle traffic counts and building heating sources, which has been previously done in creating air pollution dispersion mapping using the same techniques [64].

The inclusion of these sectors into the ETS would follow a similar path as the inclusion of the industrial sector except that the fixed amount of emission credits would be allocated to each city or port rather than an individual vehicle or building owner. The zone would then find the most efficient way to mitigate and manage risk.

This standardized approach is entirely dependent on the AQM infrastructure, which is regulated and tied to the long-standing CEN standards (1996). Allocating the responsibility for managing this risk to the city or port also helps reduce the cross jurisdictional regulation risk of the transport sector, which is currently a hodgepodge of city, country, and EU level regulation. As seen previously in the energy sector, these duplicative jurisdictions can lead to policies working against each other, resulting in zero added benefit at a greater cost. The decentralization of the decision-making to the local level minimizes risks and increases the political acceptability as this approach enables different cities to take approaches that best fit their needs, e.g., city planning.

Inclusion of the transport sector in this manner is an attractive alternative to the proposed European Carbon Border Tax as it is not a tax but a market price based on usage. This means that it is not a violation of existing international trade and diplomatic agreements. Lastly, this approach is very inexpensive with the overall cost to maintain this system being less than the implementation costs for the majority of Low Emission Zones (LEZ) [65].

4.3.7. Pricing Urban Air Pollution

Most congestion road pricing schemes are based on a flat tariff applied to vehicles entering and leaving the congestion zone, which is a fixed area at the centre of a city. Commuters tend to resist congestion schemes since they are seen as merely another tax, which many commuters believe they have already paid in the form of time along with vehicle and fuel taxes. The schemes are also viewed unfavourably on a socio-economic basis and are subject to gaming, e.g., a diesel-powered Uber vehicle that drives all day only within the zone.

However, new technologies have proven that a path-dependent road usage price can be established per vehicle as detailed by the centre for London think tank [66], which concluded that dynamic road pricing schemes are likely the most sensible and economic solutions for large cities facing intractable issues of road congestion. This approach can also be applied to pricing location-specific air pollution levels along the route travelled. Establishing this link to air pollution pricing allows cities, for the first time, to accurately place a value on clean air and improve the acceptability of a market-based dynamic congestion/air pollution scheme.

A market-based approach differs from a tax-like fixed tariff approach because a portion of the funds collected would be redistributed directly to vehicle owners or measures that reduce air pollution. For instance, a parking garage owner would be reimbursed through the market mechanism for an air purification system [17]. This could fund open air cleaning

solutions, such as 'Smog Free Towers' (https://www.studioroosegaarde.net/project/smog-free-tower (accessed on 2 May 2021)).

The overall baseline for the market price would be set by a revenue neutral entity. This is similar to what exists in the power sector with an independent system operator (ISO) who oversees the pricing mechanism for wholesale electricity. This ISO is revenue neutral with their operating budget being funded by a fixed fee based on the rent collected. These types of market structures have a long and successful record of delivering accurately priced and reliable energy. The annual baseline for this market could be raised or lowered, as dictated by the latest understanding of the health and economic costs of air pollution.

An added benefit of this approach is that baselines could be determined at a local level and are more efficient than strict vehicle bans. In countries with emissions pricing, much of the potential legal opposition has already been litigated and implementing this approach would not require unity on a national or supranational level.

## 5. Conclusions

Historically, actions taken to handle air pollution have been achieved with overwhelming political support when that pollution was visible. Making air pollution visible through the digitization of the environment at a personal level offers one of the greatest opportunities for addressing urban air pollution. This is achieved by including LCS devices in the existing environmental monitoring network via a systematic, dynamic field calibration process. The resulting, unprecedented spatiotemporal resolution of urban air quality will simplify a complex problem and empower individuals to take personal action. This greater transparency and awareness provided by the expanded air quality network will lead to consensus-driven solutions that will support numerous new, sustainable business opportunities, such as road pricing schemes linked to a market price for clean air.

Accurately measuring air pollution with a hyperlocal spatial and temporal resolution will ultimately lead to business models that are funded by sectors of the economy that benefit the most, e.g., the healthcare system. This funding may occur indirectly, such as funding from the government's transport budget for traffic management via flexible LEZs or reimbursement of the air purification system in parking garages via a market mechanism. The specific market mechanisms and mitigations used to address air pollution problems are somewhat unknown since they depend on the acceptability of the local community. However, what is clear is that we cannot begin to address this situation until we measure it.

**Author Contributions:** Conceptualization, K.S. and M.B.; methodology, K.L.; writing—original draft preparation, K.L., K.S., H.G., G.J., R.G., V.Z., M.B., and M.H.; writing—review and editing, K.S., K.L., and G.J.; project administration, K.S. and M.B.; funding acquisition, K.S. and M.B. All authors have read and agreed to the published version of the manuscript.

**Funding:** This research was partially funded by the German Federal Ministry of Traffic and Digital Infrastructure (BMVI) as part of project Smart Air Quality Network [67], https://www.smartaq.net/ (accessed on 2 May 2021), grant 19F2003A-F.

**Institutional Review Board Statement:** Not applicable.

**Informed Consent Statement:** Not applicable.

**Data Availability Statement:** Not applicable.

**Conflicts of Interest:** The authors declare no conflict of interest. The funders had no role in the design of the study, in the collection, analyses, or interpretation of data, in the writing of the manuscript, or in the decision to publish the results.

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
