# Peer review of "High-Resolution Assessment of Air Quality in Urban Areas—A Business Model Perspective"

_atmosphere, doi:10.3390/atmos12050595_

Round 1

Reviewer 1 Report

The authors have addressed the sugegsted comments well.

I'd suggest some minor revisons. In particular, I’d suggest to reduce the number of sub-sections. Many of these are too short. Often, they make hard the reading

For example, I think that Section 2.1 is not necessary. The same for the sub-section 2.2.1

I’d suggest to delete the Sub-section 3.1.1 and 3.1.2. I’d simply introduce what the reader will read in the following.

At line 158, I’d cleary specify that the existing integrated AQ networks, described above, are only some of real cases though very indicative

I think that Figure 1 has to be properly described. The concept to be shown by this figure is not clearly discussed. Perhaps, the description of the figure 1 by its caption in not sufficient for specify the concept. I’d suggest to describe properly it in the text

The Conclusions have to be better discussed. They are too generic.

In my opinion, the paper can be accepted after minor revisions

Author Response

“The authors have addressed the suggested comments well.”

Thank you for this statement.

“I'd suggest some minor revisions. In particular, I’d suggest to reduce the number of sub-sections. Many of these are too short. Often, they make hard the reading.”

We agree with this statement.

“For example, I think that Section 2.1 is not necessary. The same for the sub-section 2.2.1. I’d suggest to delete the Sub-section 3.1.1 and 3.1.2. I’d simply introduce what the reader will read in the following.”

We agree.

We deleted the headlines 2.1 Sensor Sales, and 2.2.1 sensor.community.

Line 101 – Added ‘Indicative measures’ which helps tie the skipped sections to the real-world examples.

Line 152 – Added adjective ‘technological’ to help differentiate changes in technology and changes in the way these are increasingly integrated.

Line 154 – Added adjective ‘consumer’ as a more clear reference to the previous sales paragraph.

“At line 158, I’d clearly specify that the existing integrated AQ networks, described above, are only some of real cases though very indicative.”

The above was reworded to emphasize the fact that these are ongoing real-world examples and that the result is an improved indicative air quality monitoring network. The focus on indicative measurements is a useful theme to further clarify and unify the paper. The fact that these are only select examples is implied through the rephrasing of ’types of integration‘ in the sentence which hopefully achieves this for the reader.

Line 167 – this was amended at the suggestion of Ron Evans at the EPA ([email protected]).

We deleted the headlines 3.1.1 Assessing the willingness to pay for actionable air quality data, and 3.1.2 Breathe London as well as changed the numbering of the other sections correspondingly. We introduced these paragraphs without headline by an additional sentence if necessary. For former headline 3.1.1 we added here this sentence: ‘Another topic for financing the costs of clean air is the willingness of people to pay for air quality data.’

“I think that Figure 1 has to be properly described. The concept to be shown by this figure is not clearly discussed. Perhaps, the description of the figure 1 by its caption in not sufficient for specify the concept. I’d suggest to describe properly it in the text.”

We improved and specified the description of the concept by additional text: ‘The rationale for such a self-funding mechanism is as follows: Initially, policy- or technology-based intervention and mitigation strategies (as designed and implemented by (local) governments and funded from public sources (€1)), result in improvements of ambient AQ. As a consequence of appropriate governance-based actions (Figure 1; grey-shaded box), a positive effect on public health is anticipated, which in turn results in reductions of health care costs. Resources that are saved as a result of lower health care spending can be reassigned (€2) to intensify air pollution mitigation efforts and to expand monitoring networks. In turn, this results in additional AQ improvement, leading to further exposure reduction, and consequently additional health gain. Thus, an adequate governance framework may establish proper conditions to generate a self-sustainable ‘positive feedback loop’. In this scheme, AQM serves multiple purposes: Improved public awareness may prompt individuals to use actual measurement data in making informed decisions regarding personal exposure-evasion strategies. Additionally, the availability of current and past AQ information will greatly assist in determining causality, drawing up legislation, and implementation of enforcement.’

“The Conclusions have to be better discussed. They are too generic.”

The conclusions are improved by additional sentences: ‘This greater transparency and awareness provided by the expanded air quality network will lead to consensus driven solutions that will support numerous new sustainable business opportunities such as road pricing schemes linked to a market price for clean air. Accurately measuring air pollution with a hyperlocal spatial and temporal resolution will ultimately lead to business models that are funded by sectors of the economy that benefit the most e.g., the healthcare system. This funding may occur indirectly such as funding from the government’s transport budget for traffic management via flexible LEZs or reimbursement of air purification system in parking garages via a market mechanism. The specific market mechanisms and mitigations used to address the problem of air pollution are somewhat unknown as they depend on the acceptability of the local community, but what is clear is that we can’t begin to address this situation until we measure it.’

Reviewer 2 Report

This version of the manuscript it much improved. The focus is more clear and text sticks to one organizational scheme.

Author Response

“This version of the manuscript it much improved. The focus is more clear and text sticks to one organizational scheme.”

Thank you for this statement.

Reviewer 3 Report

The manuscript is organised in five sections providing comprehensive information about the existing low cost sensors networks all around the world. While reading the manuscript I couldn’t find the description of the proposed (by the authors) business model or the smart air quality network and the contribution of the authors to the design of such a business model. In section 4 I would expect an image, diagram of the proposed calibration scheme, detailed explanations of the functionality of the Smart Air quality Network and of course the preliminary results. It seems that the authors provide the existing state of the art (comprehensively) and propose a calibration process but the manuscript lacks the results from the operation of this network and the calibration process.  I would also expect the description of the establishment of such a network (e.g. lessons learnt, obstacles, suggestions) and at least the preliminary results from the operation of the low cost sensors. Moreover, the concluding remarks are not adequately addresses. 

Author Response

“The manuscript is organised in five sections providing comprehensive information about the existing low-cost sensors networks all around the world. While reading the manuscript I couldn’t find the description of the proposed (by the authors) business model or the smart air quality network and the contribution of the authors to the design of such a business model.”

Line 314 – In order to more clearly define the proposed network, the document has been amended to emphasize that we are calling for a standardized approach to integrating IoT connected LCS devices.

In terms of the proposed business model, the funding is described in Section 4.2 Structuring and Funding the Smart Air Quality Network while the proposed business models are described in Sections 4.2.1 Mobile Network Operator; 4.3.2 Traffic Management: Flexible LEZs; 4.3.6 European Emissions Trading Scheme; and 4.3.7 Pricing Urban Air Pollution.

Line 576 - The basis for the business model has been further clarified in the Section 5 Conclusions.

“In section 4 I would expect an image, diagram of the proposed calibration scheme, detailed explanations of the functionality of the Smart Air quality Network and of course the preliminary results.”

Line 351 – We have added a paragraph to help address the lessons learned and more information on performance standards: ‘The Opensense project demonstrated that a network of mobile sensors can be consistently and reliably maintained remotely with very high levels of availability (>99%). While there is very little information available on performance comparisons using non-stationary sensor networks, this project demonstrated accuracy levels consistently above the existing indicative thresholds. In a comprehensive review of LCS performance based on stationary deployments conducted in 2019, sensors built using the same design and calibration techniques developed by Opensense were found to be the most accurate in the market at that time [9].’

Line 359 – The sentence ‘The future Smart Air Quality Network should be based on a similar dynamic calibration technique’ is to make it clearer that this technique can be further developed but the process has been developed and the approach used is the workable model to implement going forward.

Line 376 – In lieu of a calibration scheme diagram, we have included a graphical representation of the value of a network as Figure 2. In this case, it is a fusion of the Metcalf’s law and the relative proximity of those field calibration measures (both time and space) to the AQM.

In Section 4.1 The Smart Air Quality Network Design it is described: network of LCS devices in an outdoor setting; LCS devices on a mobile basis; dynamic recalibration process of fixed and mobile devices by their regular passes in close proximity and on the basis of the official air quality reference stations; setting-up three additional significant benefits of this approach; examples are given from projects and cities with references.

“It seems that the authors provide the existing state of the art (comprehensively) and propose a calibration process but the manuscript lacks the results from the operation of this network and the calibration process.”

In Section 4.1 The Smart Air Quality Network Design too these remarks and due to reduction of the manuscript length the corresponding references are given. Given its complexity, a detailed description of the calibration process would take too long but we have highlighted the performance achievements so that those interested can look up those references.

“I would also expect the description of the establishment of such a network (e.g., lessons learnt, obstacles, suggestions) and at least the preliminary results from the operation of the low-cost sensors.”

Please see information to Line 351 above.

The establishment of Smart Air Quality Networks is described in Section 2.1 Standalone, independently-operated LCS air quality assessment networks. Preliminary results from the operation of the low-cost sensors are given in Sections 2.2 Semi-integrated LCS-official air quality assessment networks for the cases; 2.2.1 AirNow: Fire and Smoke Map; 2.2.2 Cangzhou, China; and 2.2.3 Rijnmond: We-Nose Netwerk.

“Moreover, the concluding remarks are not adequately addressed.”

The conclusions are improved by additional sentences as described for review 1.

Round 2

Reviewer 1 Report

I accept in present form

This manuscript is a resubmission of an earlier submission. The following is a list of the peer review reports and author responses from that submission.

Round 1

Reviewer 1 Report

The paper objective is very interesting and relevant to regulate the use of LCS networks for high resolution air quality monitoring. However, the paper has serious weaknesses. It is not well structured. The methodology is not clearly specified.  The paper lacks a scientific evidence. The presentation quality is very bad.  The paper text contains too many web page links making heavy reading. The paper appears as a mere list of LCS businesses as well as applications or other items, the description of which is often not consistent with the specific scope of the section.  The reader has to infer the topics from these lists (see for example the Background Section). The methods, the results and the conclusions are not clearly described, discussed and addressed.

As regards the large number of web page references clearly included in the paper text, I’d suggest to create a website references section, if it is possible. For the businesses and projects examples listed in the Introduction Section, I’d suggest to create two tables and mostly to select the more representative and recent business and projects.

Then, I’d suggest to check that the referred web pages are available. Anyway, it would be preferable not to refer only to online documents.  

I think that the research has to be completely redesigned on scientific basis, rigorous methods and concepts.  

Reading this paper has been very heavy.

Reviewer 2 Report

The title of the paper points to an important point: what is a successful business model for long-term funding of low-cost air quality sensor networks? And Figure 1 presents an interesting model: the long-term costs would be recouped from savings from healthcare expenditures. I believe that a focused paper that quickly and succinctly reviewed low-cost sensor networks and then made this business model case would be important and of interest to the community of researchers that study air pollution.

That said, the current manuscript sprawls too much to achieve this aim. There are numerous tangents into many different aspects of LCS networks: calibration, integration into existing networks, etc. While it's important to have a good background, the current manuscript is too long, not well organized and not focused on the main thesis. 

Here are some general points:

  • The use of long bullet lists can occasionally be useful (yes, I'm using one now!), but they are employed too extensively in the text.
  • There are many web references, and I found many links that were broken. This material would be better as a supplement or a companion web resource that could be dynamically updated.
  • The authors should use one structure and theme to organize the paper. There are two many statements like at lines Lines 384-387 and Line 427 that attempt to organize the paper. There should be one cohesive structure, focused on the main argument of the paper.

Here are some minor points to address:

Line 49: change to “tend to resemble”.
Line 50: Define what is meant by “indicative-level monitors.” (Note: you define this around lines 89-90, so just move up the definition.)
Lines 103-104: Define CAGR and the link does not work.
Lines 118-119: Link does not work.
Lines 110-216: better to provide as a supplement to the paper, instead of including in the main text.
Line 268: Coal instead of cole.
Lines 292-293: “guarantee” is not the right word choice here. Cannot ‘guarantee’ data reliability in this context.
Lines 294-296: While this is the goal, it’s not been clearly demonstrated that LCS networks can detect exceedances apart from the reference networks.
Lines 298-299: There is no “chapter 8.”
Lines 311-322: Since the CSN is not a low-cost sensor network, better explain why it is being discussed in this context. This is somewhat introduced on lines 316-318, but go into more depth and better discuss the analogy between the CSN and LCSs.
Lines 371-373: Maybe. Either need to cite this or modify the language to make it conjecture.
Lines 384-387: This seems repetitive. What hasn’t been covered previously in the manuscript?
Line 427: Too many different competing ways to organize the paper.

Reviewer 3 Report

Manuscript Number: atmosphere_1058569

The main scope of the article is to present different models for the application of low cost sensors (LCS) networks in urban scale. Existing methodologies (business models) for the estimation of air quality with the use of LCS are mentioned in detail. A comprehensive report of the businesses selling LCS, services for network measurements and online platforms for the visualisation of real time measurements, projects dealing with smart city networks as well as existing networks of low cost sensors.

It is a fact that many LCS networks operate in many cities worldwide, either developed by local authorities or in the frame of research projects. The sustainability of all these networks is very important since they provide useful information of air quality by covering a large extent of urban areas and operate complementarily to the official network, the utilities of which are very expensive and difficult to expand.

Generally the manuscript is well arranged in six sections and the proposals made towards the application of business models are of great interest. However, some corrections are either needed or required and clarifications should be provided in order to improve the paper to be accepted. Please read the following comments.

Point 1: lines 275-279. Please clarify who is responsible for the development of the mentioned instrument that will measure black carbon.

Point 2: lines 285 – 286. “Other reports from Europe where known from literature”. Please add references.

Point 3: lines 474 – 501. Concerning the “Smart air quality network: Full-scale Build” described in the text please explain on which part of the vehicle these utilities should be placed. The authors should mention to what extent the pollutants emitted from the public transport vehicles will affect the measured concentrations. These sensors will mostly depict the concentrations close to the road and will be affected by traffic. The background concentrations will be difficult to catch.

Point 4: lines 529 – 543. Google is not the only one that has made efforts to create an urban emissions inventory. There are many urban emission inventories referring to large cities or counties that include emissions profiles in multiple temporal scales (annual, daily, hourly). Many pollutants are covered included GHGs. Many of these emissions inventories are based on official traffic counts, population and energy consumption data and participate at JRC initiatives. The authors should mention at least some of them.

Please see the below references.

https://www.scopus.com/record/display.uri?eid=2-s2.0-85081262397&origin=resultslist

https://www.scopus.com/record/display.uri?eid=2-s2.0-85070629672&origin=resultslist

https://www.scopus.com/record/display.uri?eid=2-s2.0-84964499644&origin=resultslist

https://www.scopus.com/record/display.uri?eid=2-s2.0-84878723035&origin=resultslist

Point 5: lines 540 – 543. Please explain how the development of a city level emission inventory will be achieved given the fact that sensors measure pollutant concentrations and not emissions. There are models that provide evidence of the pollutants sources via the use of source apportionment applications but they do not calculate emissions. Do the existing low cost sensors have the ability to perform such an application and provide emissions measurements? Please mention existing building and traffic emissions (not concentrations) monitoring systems in cities as refered at the above lines. The section 4.2.2 should be rewritten and enriched with existing emissions inventories the imprtance of which should be mentioned.